# Preparation of Micro-Size Spherical Silver Particles and Their Application in Conductive Silver Paste

**DOI:** 10.3390/ma16041733

**Published:** 2023-02-20

**Authors:** Na Li, Jun Li, Xiaoxi Wan, Yifan Niu, Yongwan Gu, Guo Chen, Shaohua Ju

**Affiliations:** 1Faculty of Metallurgical and Energy Engineering, Kunming University of Science and Technology, Kunming 650093, China; 2Key Laboratory of Unconventional Metallurgy, Ministry of Education, Kunming 650093, China; 3National Local Joint Laboratory of Engineering Application of Microwave Energy and Equipment Technology, Kunming 650093, China; 4Kunming Institute of Precious Metals, Kunming 650106, China

**Keywords:** spherical silver particle, micro-size, monodispersed, resistivity

## Abstract

In this paper, micro-size spherical silver particles were prepared by using a wet-chemical reduction method. The silver particles were characterized by scanning electron microscopy (SEM), X-ray diffraction (XRD) and a laser particle-size analyzer. The results indicate that different types and the content of surfactants can be used to prevent the accumulation, and control the morphology and particle size distribution, of silver particles. Moreover, the morphology of silver particles was changed from polyhedral to spherical when the pH was raised from 1 to 3. Under the optimal synthesis conditions (0.1 mol/L silver nitrate, 0.06 mol/L ascorbic acid, gelatin (5% by weight of silver nitrate), pH = 1), the micro-size spherical silver particles with diameter of 5–8 μm were obtained. In addition, the resistivity of conductive silver paste that prepared with the as-synthesized spherical silver particles was discussed in detail and the average resistivity of the conductive silver paste was 3.57 × 10^−5^ Ω·cm after sintering at 140 °C for 30 min.

## 1. Introduction

With the rapid development of the electronic components industry, conductive silver paste is widely used in liquid crystal display (LCD), SMD light-emitting diodes (LED), integrated circuit (IC) chips, printed circuit board assemblies (PCBA), ceramic capacitors, membrane switches, smart cards, RF identification and other electronic components because of its excellent electrical conductivity, adhesion, solderability, bending resistance and simple process, precise wiring and easy cost control [1,2,3,4,5]. Metallic silver powder is the main component of conductive silver paste. Its conductive properties mainly rely on silver powder. The content of silver powder in the paste directly affects the conductive properties. Generally, micron-size silver powder has a lower surface resistance, and when applied to silver paste, it has higher vibrancy density and lower cost than nano silver powder [6].

Many methods have been developed for the preparation of silver particles, such as the direct current arc plasma method, spray thermal decomposition method, thermal decomposition method, electrolysis method, ultrasonic chemical method, precipitation conversion method, microemulsion method, mechanical ball grinding method and so on [7,8,9,10]. The wet-chemical reduction method has become the primary method for preparing silver particles because of its advantages such as ease of operation, low cost and energy-saving [11]. By optimizing the synthesis conditions, such as controlling the strength of solution, chemical properties of stabilizer, stirring method, stirring speed, reaction temperature, solution pH value and other process parameters, the morphological characteristics and related properties of particles can be adjusted [12,13,14,15,16,17].

Qin et al. [18] synthesized quasi-spherical silver nanoparticles with ascorbic acid as reductant and citrate as stabilizer in 30 °C water bath. Spherical silver particles with sizes of 30–72 nm were successfully obtained by changing the pH value of the reaction system. An et al. [19] has proposed a new wet chemistry method for the large-scale preparation of ultra-fine homogeneous spherical silver powder with ascorbic acid as reductant and citrate as stabilizer. It was found that the reaction temperature has an important effect on the particle size. The mean diameter of the particles decreased from 3.5 μm to 1.6 μm as the reaction temperature increased from 8 to 15 °C. Xie et al. [20] used PVP as dispersant, ascorbic acid as reductant and AgNO_3_ as silver source to prepare highly dispersed silver micro/nanoparticles. It was found that PVP/AgNO_3_ mass ratio could significantly alter the particle shape and size, while pH and temperature were the main factors influencing the particle size. A possible mechanism for the preparation of highly dispersed silver nanoparticles for shape-controlled synthesis is that PVP is applied to the surface of the particles to make them grow evenly on each surface. At 40 °C, PVP/AgNO_3_ mass ratio of 0.6 and pH of 7, the mean particle size of silver powder is 0.2 μm. Tan et al. [21] studied the effect of silver ion concentration on the particle size and morphology of ultrafine silver powder and found that the particle size of silver powder increased with the increasing of silver ion concentration. Still, there was no significant change in the morphology of silver powder as the concentration of silver ions increased. This is because the increase of silver ion concentration accelerates the reduction reaction rate, and there is enough silver ion in the system to grow crystal nuclei. Hence, the silver powder particle size is large.

Generally speaking, the electrical conductivity of silver particles can be adjusted by controlling the size, shape, particle size distribution and crystallinity [22]. The regular particle shape and uniform particle size distribution are conducive to the closer filling of silver powder particles in conductive silver paste, so as to form more conductive pathways and improve electrical conductivity. Therefore, studying the preparation and application of micron silver particles is of great significance. In this paper, we present a method for preparing micro-size spherical silver particles. Silver nitrate was used as the silver source and ascorbic acid was used as reducing agent. The influence of different dispersants, the solution pH and the stirring speed on the shape and size of the spherical silver particles were investigated. The effects of sintering temperature and sintering time on the conductivity of the silver paste are discussed.

## 2. Materials and Methods

### 2.1. Materials

All materials used in this study are shown in Table 1, and ultrapure water was used in all of the experimental processes.

### 2.2. Preparation of Silver Particles

Figure 1 shows the experimental device for the synthesis of spherical silver particles. The two reactant solutions were prepared as follows: 0.1 mol/L silver nitrate solution was prepared by dissolving 1.7 g silver nitrate in 100 mL ultrapure water, and the pH was adjusted by nitric acid, then stirred to obtain solution 1. The 0.06 mol/L ascorbic acid solution was prepared by dissolved 1.056 g ascorbic acid in 100 mL ultrapure water, then 0.085 g of surfactant (5% by weight of silver nitrate) was added to the solution and stirred evenly to obtain reduction solution 2.

Figure 2 shows the synthetic process and conditions for preparation of monodispersed silver particles. Solution 2 was quickly added to solution 1 and the mixed solution was stirred at 25 °C (±3 °C) for 30 min. After the stirring was stopped, the solution was precipitated layer by layer. The reacted solution was filtered, the precipitate was washed with ultrapure water and then the precipitate was dried to obtain silver powder.

### 2.3. Characterization

The surface morphologies of the silver particles were detected using a scanning electron microscope (SEM, JSM-6610A, Tokyo, Japan) operating at an acceleration voltage of 20 kV. The crystal structure of the silver particles was described by X-ray diffraction (XRD, XPert Powder, the Netherlands) with a scanning speed of 8° min^−1^ from 10 to 90°. The particle size distribution of silver powders was detected by a laser particle sizer (Rise-2002, Jinan). The amount of organic residue on the silver particle surface was measured by thermogravimetric analysis (TG, Mettler TGA2, Bern, Switzerland) and temperature rate of 30 to 800 °C using a flow rate of gas is 20 °C/min.

The conductive silver paste was prepared by mixing 0.8 g synthesized silver sample and 0.2 g organic carrier evenly in the agate. Subsequently, the prepared paste was uniformly printed on a polyethylene terephthalate (PET) substrate with screen-printing mesh, to form different shapes of conductive films (100 mm × 10 mm, 1500 mm × 0.3 mm and 1500 mm × 0.4 mm). Then the conductive film was sintered at different temperatures (100–180 °C) and times (5–45 min), and the resistivity of the sintered film was measured by a four-probe instrument.

The calculation formula of the resistivity is:(1)ρ=RS/L
where ρ (Ω·cm), *R* (Ω), *S* (cm^2^) and *L* (cm) represent the resistivity, surface resistance, cross-sectional area and length of the cured silver paste.

## 3. Results and Discussion

### 3.1. Characterization of the Synthetic Silver Particles

The XRD spectrum of the particles prepared with different dispersants is exhibited in Figure 3. The diffraction peaks have five characteristics, corresponding to the crystal faces of (111), (200), (220), (311) and (222) of face-centered cubic silver, which are consistent with the pattern of standard crystal silver card (JCPDS#04-0783). It means that the prepared sample is metallic silver. It shows that the use of different dispersants does not affect the composition of silver powder. The intensity of the peaks reflects the high crystallinity of the silver nanoparticles, and it can be seen from Figure 3 that the silver powder prepared with gelatin as dispersant has the highest crystallinity, so gelatin was used as the dispersant in this experiment.

To study the thermal stability, thermogravimetric analysis was carried out. The thermal analysis of the silver powder (TGA curve) is shown in Figure 4. It shows that the curve drops sharply between 130 °C and 250 °C, which may be due to the volatilization of solvent in the silver powder. The weight loss of the whole sample was only 0.73%, indicating that there was little organic material in the silver nanoparticles.

### 3.2. Effects of Experimental Condition on Morphology and Particle Size of Silver Particles

#### 3.2.1. Effect of Dispersant

Using the reaction solution (1) and (2), when 4 g of nitric acid were added, the silver nitrate was stirred at 100 rpm in the beaker and amounts of different dispersants were 5% by weight of silver nitrate (other conditions correspond to the description of the test procedure in Section 2.2). The reaction was performed by adding the solution 2-dropwise to the solution 1 in 2 min. Silver powders made by different dispersants were characterized by SEM, as shown in Figure 5.

As can be seen from Figure 5, different dispersants have a great influence on the shape of silver powder, and the particle size distribution of silver powder formed by four dispersants is relatively concentrated. As illustrated in Figure 5a–d, when gelatin is used as dispersing agent, silver powder is a high-crystal spheroid with smooth crystal surface and good dispersion. The mean particle size is 5 to 8 μm, but when stearic acid, succinic acid and glutaric acid are used as dispersants, the results show that they are spherical in shape and appear to be highly agglomerating. It is obvious that the silver powder prepared with gelatin as a dispersant has better monodispersity than other silver powders, and the particle surface smoothness is higher. Sannohe et al. [23] also found that the polymer compound structure has an important role for silver with monodisperse particles due to the adsorption of polymer compounds on the surface of the growing silver particles.

The relationship between the D_50_ of the silver powder and the dispersant is given in Figure 6. From the results of Figure 5 and Figure 6, it can be seen that the trend of particle size distribution of silver powder is consistent with the SEM results. Therefore, the dispersant is a key factor affecting the appearance and particle size of silver powder, and the selection of suitable dispersant is very important for the synthesis of micron-size spherical silver powder.

#### 3.2.2. Effect of pH Values

According to the results in Section 3.2.1, the influence of different pH values on the shapes and particle size distribution of the spherical silver powders were studied. The pH of the silver nitrate solution was adjusted to 1.0, 2.0 and 3.0 by adding different amounts of nitric acid to solution 1. The reaction was performed in a dropwise manner with the addition of solution 2 in solution 1 for 2 min, and the other conditions correspond to the description of the test procedure in Section 2.2. The SEM image and particle size distribution of the spherical silver powder are shown in Figure 7.

From Figure 7, with the increase of pH from 1.0 to 3.0, the shape of silver particles is changed from polyhedral to partially spherical, and the mean diameter of silver particles is reduced. A possible reason is that the growth of silver powder is affected by the solution’s pH value, which directly affects the ionization degree of ascorbic acid in it. This results in different driving forces for the reduction of silver powder.

The half-reaction of ascorbic acid is C_6_H_6_O_6_ + 2H ^+^ +2e = C_6_H_8_O_6_. The reduction electrode potential of C_6_H_8_O_6_ can be calculated by the following equation: E = E^0^ − 0.059 pH [24]. Therefore, the reducing ability of ascorbic acid can be adjusted by changing the pH value, and the reducing ability of ascorbic acid increases with the increase of pH value. When the pH value is low, the reduced ability of C_6_H_8_O_6_ is relatively low, and only a tiny amount of silver nuclei is generated in the solution. The silver nuclei formed grow in a specific crystal direction, leading to the formation of polyhedron structure [25,26,27]. With the increase of solution pH value, the reduced ability of C_6_H_8_O_6_ solution increases, and more silver nuclei will be formed. The increase in the number of silver nuclei will limit the growth of crystal nuclei, leading to the reduction of silver particle size.

#### 3.2.3. Effect of Stirring Speed

The effect of stirring speed on particle shapes and particle size distribution of spherical silver powders was investigated. The pH of solution 1 was adjusted to 1 and solution 2 was protected with 5% gelatin by weight of silver nitrate. Solution 2 was added dropwise to solution 1 for reaction, and the speed of the reaction solution was adjusted to 50 rpm, 100 rpm, and 200 rpm, the other conditions correspond to the description of the test procedure in Section 2.2.

The SEM images and the particle size distribution of the spherical silver powder at different stirring speeds were presented in Figure 8. From the SEM images a to c, it can be seen that the stirring speed significantly influences the shape of the silver powder. As illustrated in Figure 8, when the stirring speed was 50 rpm, the silver powder was polyhedral and spherical, and its surface was smooth. When the speed of stirring was increased to 100 rpm, there were obvious large and small particles, and the distribution of silver particles was uneven. This may be caused by the agglomerative growth of small particles. In addition, when the stirring speed was increased to 200 rpm, the smaller particles gradually disappeared, and the particle size distribution of the silver particles became more uniform. In this case, the particle size distribution of the silver particles became wider, and the dispersibility became lower. This is probably due to the partial destruction of the interaction between the dispersing agent and the silver particles, which results in the formation of new aggregates at high stirring speed.

### 3.3. Possible Formation Mechanism of Spherical Silver Particles

This study used nitric acid as a pH regulator and gelatin as a stabilizer and dispersant. The results show that the type of dispersant, solution pH and stirring speed play an essential role in the formation and growth of silver particles, which may affect the morphology or distribution of silver particles. Figure 9 shows the mechanism of the growth process of spherical silver particles. Firstly, some Ag^+^ nucleates on the surface of polyhedral spherical silver particles grow and penetrate polyhedral, spherical silver particles under the action of diffusion. However, the rest of the Ag^+^ nucleates uniformly and contacts the first nucleated silver particles to form aggregates. By aggregate growth, the primary particles will further grow into secondary particles [28]. Due to the adsorption of gelatin on the surface of silver particles, the long chain structure of the gelatin can provide a good space constraint, thereby preventing the aggregation of silver particles and the formation of secondary particles [29]. Consequently, the silver particles are well dispersed, and it is possible to control particle size.

### 3.4. Resistivity of Silver Paste

The silver particles used in the preparation of conductive silver paste were carried out under the following conditions: silver nitrate 0.1 mol/L, ascorbic acid 0.06 mol/L, the content of gelatin was 5% by weight of silver nitrate, and pH = 1.0, and the speed of the reaction solution was adjusted to 50 rpm. The conductive silver paste was prepared according to the description of the test process in 2.3 and printed on polyethylene terephthalate (PET) substrate to form different shapes of conductive films (100 mm × 10 mm, 1500 mm × 0.3 mm and 1500 mm × 0.4 mm). The screen-printing mesh was shown in Figure 10a, and the printed conductive films were shown in Figure 10b.

Measure the thickness of the conductive film using a micrometer (thickness of the conductive film printed on the PET substrate minus the thickness of the PET substrate). Using the resistance value measured by the four probes and the thickness information measured by the micrometer, the resistivity of the conductive film can be calculated. The thickness of five different points is measured on a conductive film pattern, the average of which is used to calculate resistivity. The average resistivity of three different conductive film patterns is shown in Table 2. The average resistivity of the three conductive films was calculated according to Equation (1) in Section 2.3, and the results are shown in Figure 11. The error bar in Figure 11 was based on the standard.

The sintering process of the conductive silver paste has an extremely important impact on its conductivity. The proper sintering temperature and sintering time are conducive to the formation of a low resistivity conductive silver film [30,31]. Figure 11a shows the average resistivity of the three conductive films versus different temperatures (100–180 °C) for 10 min. It can be clearly seen that the resistivity decreased slowly with the increasing of sintering temperature. When the temperature is 100 °C, the average resistivity is 7.40 × 10^−5^ Ω·cm. Increased the sintering temperature to 140 °C, the average resistivity drops to 3.57 × 10^−5^ Ω·cm. However, with the sintering temperature increased to 140–180 °C, the average resistivity changes slowly from 3.57 × 10^−5^ Ω·cm to 3.11 × 10^−5^ Ω·cm. This is mainly because most of the solvent in the silver film has evaporated and the silver particles were effectively connected with the shrinkage of the film. Only a small portion of the solvent is volatilized by subsequent reheating, which has little effect on the electrical properties.

Figure 11b shows the average resistivity of the three conductive films versus different sintering times (5–45 min) at 140 °C. It shows that the average resistivity changed little with the increased sintering time, and the average resistivity is 3.88 × 10^−5^ Ω·cm after sintering at 140 °C for 5 min. When the sintering time increased to 30 min, the resistivity dropped a little to 3.57 × 10^−5^ Ω·cm, and when the sintering time was increased to 35–45 min, the average resistivity dropped slowly from 3.47 × 10^−5^ Ω·cm to 3.44 × 10^−5^ Ω·cm. The possible reason is due to the silver film being almost sintered completely when sintered at 140 °C for 5 min and could not be further reduced substantially by prolonging the sintering time.

It had been reported that a silver paste consisting of 80 wt.% Ag nanoparticles, 1.0 wt.% lead-free glass material and 19 wt.% organic carrier have a volume resistivity of 4.11 × 10^−6^ Ω·cm after sintering at temperatures ranging from 250 °C to 450 °C [32]. However, the conductive silver paste we studied has a lower sintering temperature (140 °C).

## 4. Conclusions

A simple, efficient and fast method for the preparation of micron-size spherical silver powder was presented in this paper. Silver nitrate as precursor, gelatin as dispersant and ascorbic acid as reducing agent were used to produce silver powder with an average particle size of 5~8 μm. The results indicate that the type of dispersant, the pH value of the solution and the stirring speed have significant effects on the powder’s morphology and particle size distribution of silver powder. The conductive silver paste produced by the prepared micron-size spherical silver powder has an excellent electrical conductivity of 3.57 × 10^−5^ Ω·cm after sintering at 140 °C for 30 min, which indicates that the prepared micron-size spherical silver powders can be used as a conductive silver paste.

## Figures and Tables

**Figure 1 materials-16-01733-f001:**
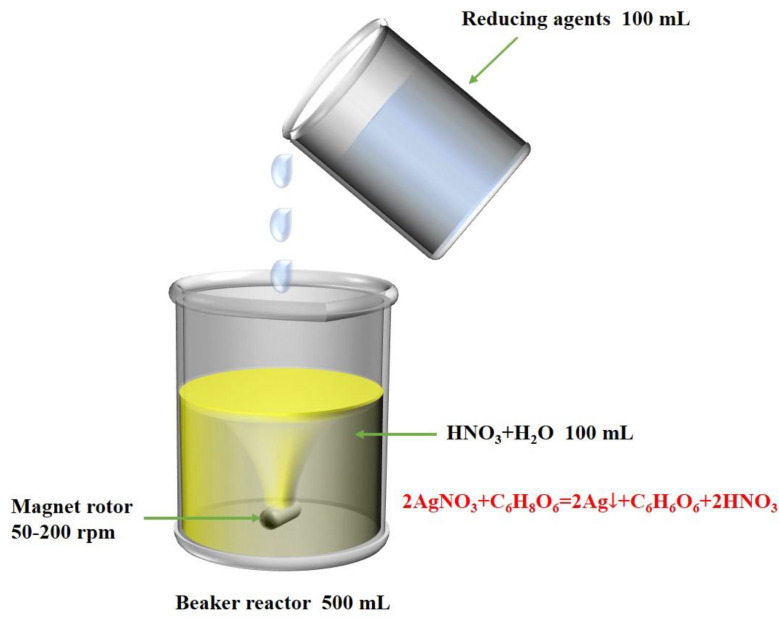
Experimental device for silver particle synthesis.

**Figure 2 materials-16-01733-f002:**
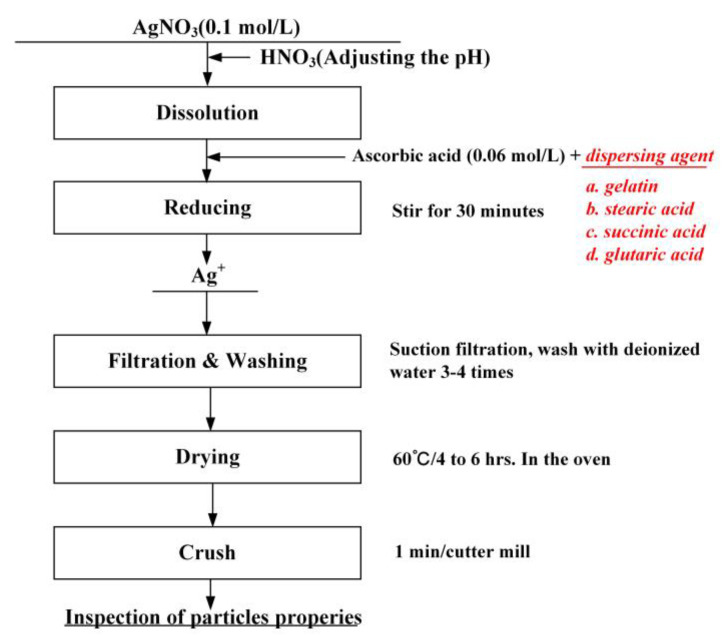
Synthetic flow chart for preparation of monodispersed silver particles.

**Figure 3 materials-16-01733-f003:**
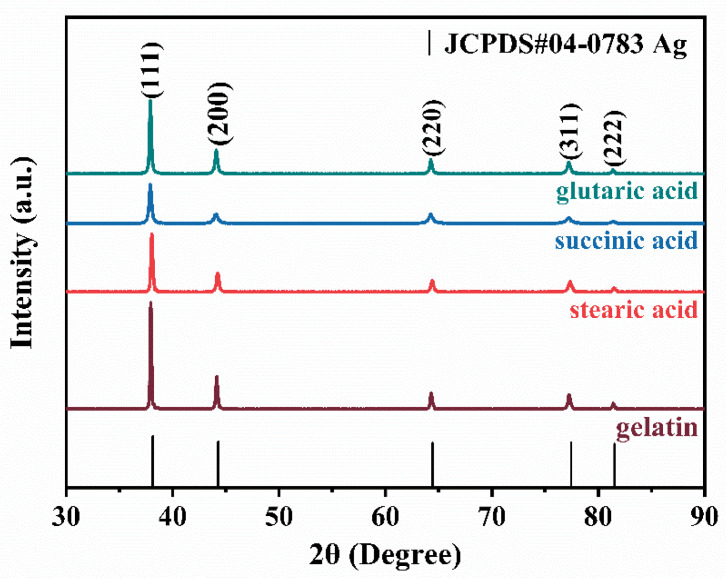
XRD of silver powder prepared with different dispersants.

**Figure 4 materials-16-01733-f004:**
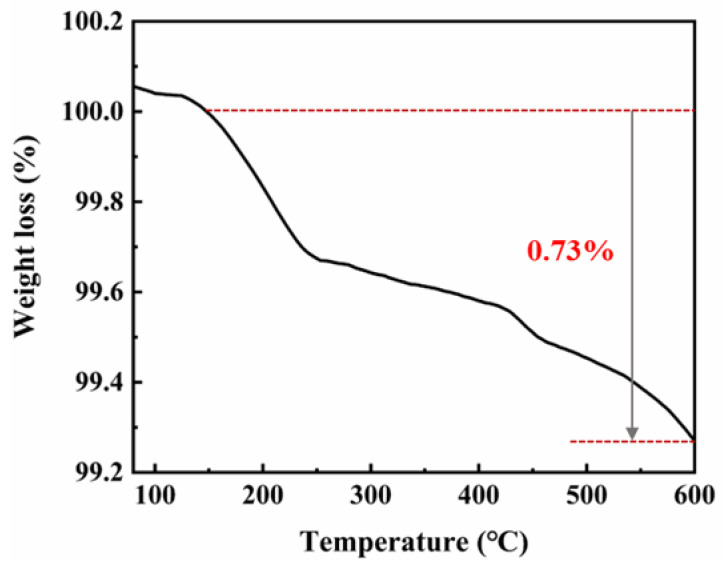
TGA thermogram of silver powder.

**Figure 5 materials-16-01733-f005:**
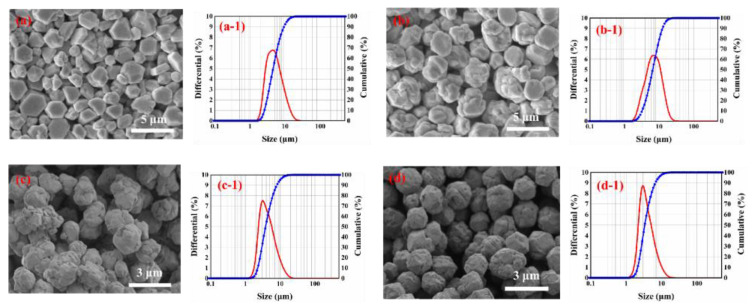
SEM images of silver particles prepared with different dispersants: (**a**) gelatin; (**b**) stearic acid; (**c**) succinic acid; (**d**) glutaric acid. Size distribution of silver particles prepared with different dispersants: (a-1) gelatin; (b-1) stearic acid; (c-1) succinic acid.

**Figure 6 materials-16-01733-f006:**
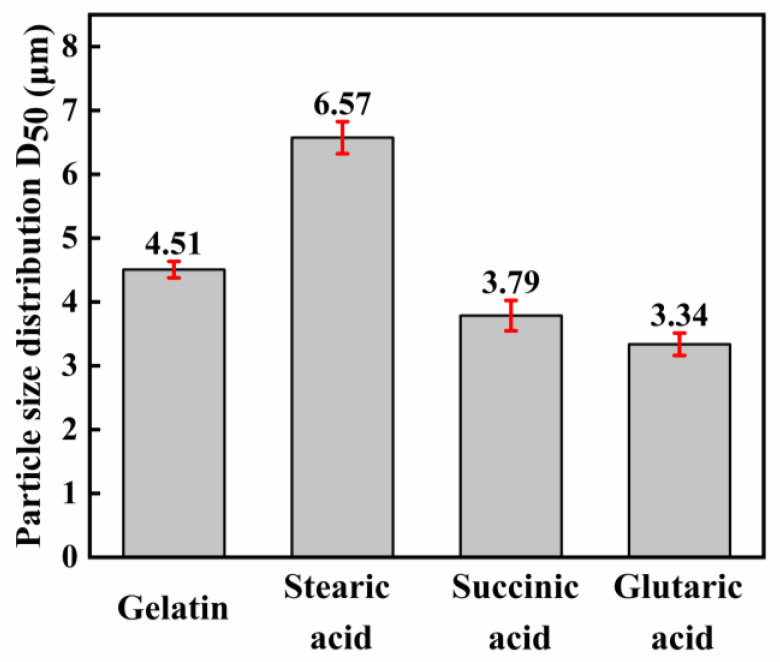
Variation of the average size of silver particles (D_50_) with different dispersants.

**Figure 7 materials-16-01733-f007:**
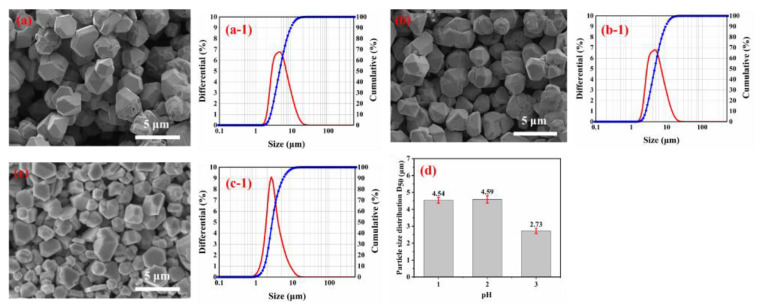
SEM images of silver particles prepared with different pH values: (**a**) pH = 1.0; (**b**) pH = 2.0; (**c**) pH = 3.0; (**d**) Variation of the average size of silver particles (D_50_) with pH value. Size distribution of silver particles prepared with different pH values: (a-1) pH = 1.0; (b-1) pH = 2.0; (c-1) pH = 3.0.

**Figure 8 materials-16-01733-f008:**
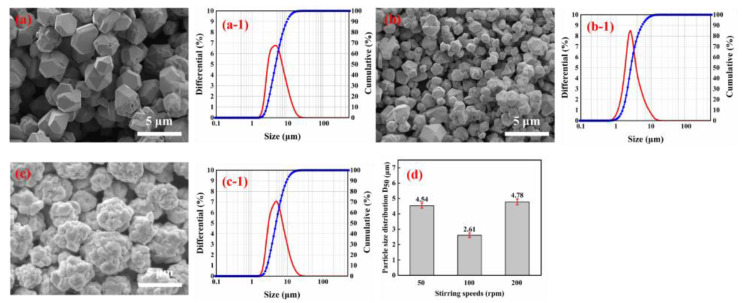
SEM images of silver particles prepared with different stirring speeds: (**a**) rpm = 50; (**b**) rpm = 100; (**c**) rpm = 200; (**d**) Variation of the average size of silver particles (D_50_) with stirring speeds. Size distribution of silver particles prepared with different stirring speeds: (a-1) rpm = 50; (b-1) rpm = 100; (c-1) rpm = 200.

**Figure 9 materials-16-01733-f009:**
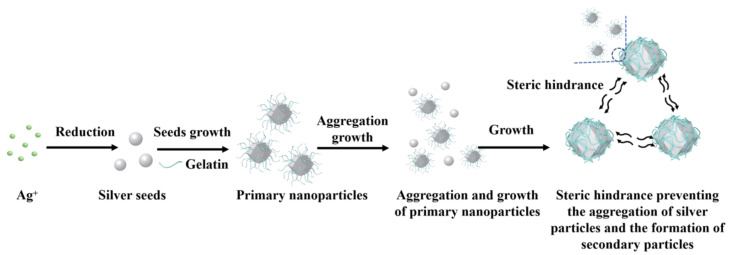
A schematic illustration of the proposed growth process of spherical micrometer silver particles.

**Figure 10 materials-16-01733-f010:**
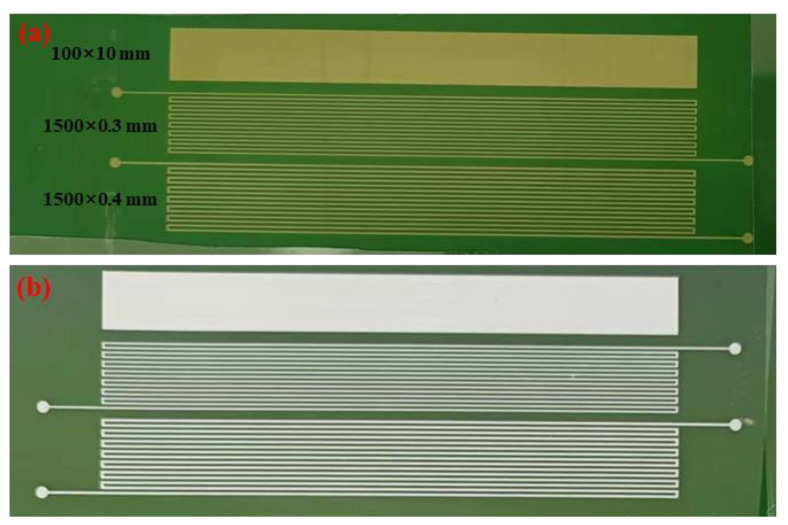
The screen-printing mesh (**a**) and the printed conductive films (**b**).

**Figure 11 materials-16-01733-f011:**
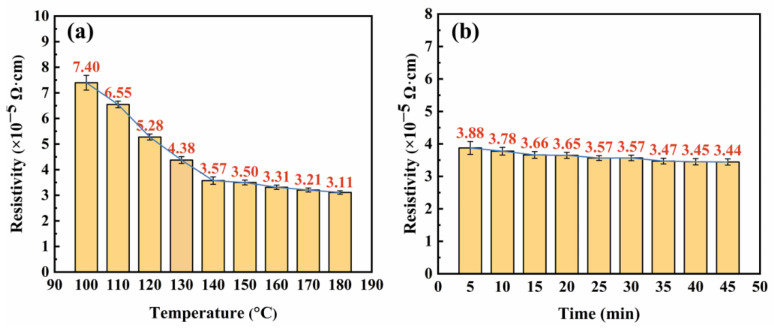
The average resistivity of conductive films: (**a**) Resistivity versus sintering temperature (**b**) Resistivity versus sintering time.

**Table 1 materials-16-01733-t001:** Materials used in this study.

Materials	Chemical Formula	Specification
Silver nitrate	AgNO_3_	≥99.8%
Ascorbic acid	C_6_H_8_O_6_	≥99.7%
Nitric acid	HNO_3_	98%
Absolute ethanol	C_2_H_6_O	95%
Gelatin	—	Industrial-grade
Stearic acid	C_18_H_36_O_2_	Analytical Reagent
succinic acid	C_4_H_6_O_4_	Analytical Reagent
glutaric acid	C_5_H_8_O_4_	Analytical Reagent

**Table 2 materials-16-01733-t002:** The average resistivity of three different conductive film patterns.

Notations	Length (mm)	Width (mm)	Thickness (μm)	Resistance (Ω)	Resistivity (Ω·cm)
Line 1	100	10	6.7	5.42	3.63 × 10^−5^
Line 2	1500	0.3	6.3	2777.78	3.58 × 10^−5^
Line 3	1500	0.4	5.6	2 343.75	3.50 × 10^−5^

## Data Availability

The data presented in this study are available in the article.

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
