# Peer review of "Preparation of Micro-Size Spherical Silver Particles and Their Application in Conductive Silver Paste"

_materials, 2023, doi:10.3390/ma16041733_

Round 1

Reviewer 1 Report

Error bars in figures, starting from Fig.4, should be added and meaningful digits in the measured numbers should be left. Minor corrections - e.g., Ref.12 (Kon'Kova - change to Kon'kova) should be done prior acceptance.

Reviewer 2 Report

The reviewed article by Ju et al. deals with the preparation of micro-size spherical silver particles and their application in conductive silver paste. The research topic undertaken is interesting and is currently arousing wide interest in the scientific world. However, in its present form, the article does not show the Authors' contribution to the existing state of knowledge on this topic.

The Authors, in my opinion, should clearly emphasize what constitutes an element of novelty in their work, since research on obtaining Ag nanoparticles in the presence of ascorbic acid as a reducing agent and gelatin as a dispersing agent has already been carried out for many years and the number of publications on this topic is considerable.

I believe that the work needs to be deeply revised and supplemented, because in its current form it cannot be published in the Materials journal.

Details of the review are shown in the attached file.

Reviewer 3 Report

This study is systematic and interesting, although the subject matter is not new. This work can be considered as promising for publication if the authors:

1) Focus in detail on the explanation of the facts obtained;

2) More carefully consider the role of pH in the formation of silver nanoparticles of a certain shape. Here it is necessary to expand the range of studied pH and to register SEM at intermediate pH values;

3) The conclusions in the article seem trivial and are less new than the work itself. It seems that it is necessary to describe in more detail the colloid-chemical regularities of the formation of silver nanoparticles;

4) The introduction and the volume of the considered original literature seem to be very small for such a comprehensively researched direction (although I do not deny some novelty of the manuscript, the introduction needs to be finalized)

Round 2

Reviewer 2 Report

The Authors addressed all the comments and clearly improved their manuscript. Therefore, I believe that now it can be published in the Materials journal. However, I still have a minor remark: in the lines 287-291 (Conclusions section) part of the sentence has been duplicated. The Authors should correct this before publication.

Reviewer 3 Report

The authors carefully revised the manuscript, now it can be accepted.
